# Quantifying the critical thickness of electron hybridization in spintronics materials

T. Pincelli[1,2,*], V. Lollobrigida[1,3,*], F. Borgatti[4], A. Regoutz[5], B. Gobaut[6], C. Schlueter[7], T.-L. Lee[7], D.J. Payne[5], M. Oura[8], K. Tamasaku[8], A.Y. Petrov[1], P. Graziosi[4], F. Miletto Granozio[9,10], M. Cavallini[4], G. Vinai[1], R. Ciprian[1], C.H. Back[11], G. Rossi[1,2], M. Taguchi[8,12], H. Daimon[12], G. van der Laan[7] & G. Panaccione[1]

In the rapidly growing field of spintronics, simultaneous control of electronic and magnetic properties is essential, and the perspective of building novel phases is directly linked to the control of tuning parameters, for example, thickness and doping. Looking at the relevant effects in interface-driven spintronics, the reduced symmetry at a surface and interface corresponds to a severe modification of the overlap of electron orbitals, that is, to a change of electron hybridization. Here we report a chemically and magnetically sensitive depth-dependent analysis of two paradigmatic systems, namely $La_{1-x}Sr_xMnO_3$ and $(Ga,Mn)As$. Supported by cluster calculations, we find a crossover between surface and bulk in the electron hybridization/correlation and we identify a spectroscopic fingerprint of bulk metallic character and ferromagnetism versus depth. The critical thickness and the gradient of hybridization are measured, setting an intrinsic limit of 3 and 10 unit cells from the surface, respectively, for $(Ga,Mn)As$ and $La_{1-x}Sr_xMnO_3$, for fully restoring bulk properties.

[1] Istituto Officina dei Materiali-CNR, Laboratorio TASC, Area Science Park, S.S. 14, Km 163.5, Trieste I-34149, Italy. [2] Dipartimento di Fisica, Università di Milano, Via Celoria 16, Milano I-20133, Italy. [3] Dipartimento di Scienze, Università degli Studi Roma Tre, Via della Vasca Navale 84, Roma I-00146, Italy. [4] Consiglio Nazionale delle Ricerche—Istituto per lo Studio dei Materiali Nanostrutturati (CNR-ISMN), via P. Gobetti 101, Bologna I-40129, Italy. [5] Department of Materials, Imperial College London, South Kensington, London SW7 2AZ, UK. [6] Sincrotrone Trieste S.C.p.A., S.S. 14 Km 163.5, Area Science Park, Trieste 34149, Italy. [7] Diamond Light Source, Harwell Science and Innovation Campus, Didcot OX11 0DE, UK. [8] RIKEN SPring-8 Center, Kouto 1-1-1, Sayo-cho, Sayo-gun, Hyogo 679-5148, Japan. [9] CNR-SPIN, Complesso Universitario Monte S. Angelo, Napoli 80126, Italy. [10] Dipartimento di Fisica, Università 'Federico II' di Napoli, Napoli, 80126, Italy. [11] Institut fur Experimentelle Physik, Universitat Regensburg, Regensburg D-93040, Germany. [12] Nara Institute of Science and Technology, 8-9165 Takayama, Ikoma, Nara 630-0192, Japan. * These authors contributed equally to this work. Correspondence and requests for materials should be addressed to G.P. (email: panaccioneg@elettra.eu).

The effectiveness of electron hybridization in solids and its competition with Coulomb interactions plays a fundamental role in novel physical phenomena, often termed as quantum properties[1,2]. In the context of spintronics, magnetic and electronic reconstructions at interfaces have been often reported, with their origin lying in the delicate interplay between charge, spin and orbital degrees of freedom[1–5]. Looking at the strength of electron hybridization and localization, near a surface or interface the reduced translational symmetry breaks or severely alters the electronic properties with important consequences for, for example, the magnetic order parameter, transition temperature and metallic/insulator character, thus potentially limiting the achievement of the desired performance in interface-based devices[6–8]. Moreover, surface- and defect states play critical roles in mediating ferromagnetism, due to the modified chemistry of the first top layers.

Prototypical spintronics systems displaying such effects are the rare-earth-doped manganites, in particular metallic La$_{1-x}$Sr$_x$MnO$_3$ (LSMO), and the most representative diluted magnetic semiconductor, (Ga,Mn)As. In both systems, the relationship between electronic reconstruction and magnetic properties and the competition between electron localization and hybridization are relevant ingredients in determining their Curie temperature ($T_C$) and ferromagnetic state[5–12]. In LSMO, the mechanism and the reason for the modified electronic properties of the surface region are still open questions; in (Ga,Mn)As a carrier depletion zone up to 1 nm has been found in the vicinity of the surface, with modified ferromagnetic order[11]. Moreover, a remarkable example of altered electronic properties has been reported in the so-called magnetic 'dead layer' at the surface of otherwise ferromagnetic bulk systems[13–17].

To date, bulk sensitive techniques, exploiting the combination of aberration-corrected transmission electron microscopy and electron energy loss spectroscopy was recently able to quantify the role of the charge-transfer screening length at the interface LSMO/PZT (lead zirconate titanate)[17], and revealed interfacial electronic reconstruction and a change in $T_C$ near the metal–insulator transition in both LSMO/STO and (Ga,Mn)As/GaAs (refs 11,18). Furthermore, surface-sensitive tools, such as angular resolved photoemission spectroscopy (PES) and scanning probes, gave clear indications of a negligible coherent spectral weight at the Fermi level in bilayer LSMO crystals, with a more fragile metallic and magnetic character at the surface than in the bulk[16,19].

Although general agreement has been reached on the observation that both metallicity and ferromagnetism of these systems are reduced at the surface, the determination of the crossover between surface and bulk properties, that is, what the 'critical' thickness of such an effect is, and whether the crossover is smooth or abrupt, needs a more complete, and preferably quantitative, description, with particular attention to the modification of the bulk electronic properties when approaching the surface.

Here we report results obtained on thin films of metallic LSMO (with $x = 0.33$ and $x = 0.35$) and of (Ga,Mn)As (with Mn doping between 8 and 13%) using core-level X-ray PES. The large tuneability of the photon energy offered by synchrotron radiation is exploited to significantly vary the information depth from the surface region ($<10$ Å, corresponding to a few atomic layers) down to the bulk ($>100$ Å; refs 20,21). We provide direct and quantitative information of the evolution from metallic (bulk) to insulating (surface) character of these materials, together with a clear indication of the behaviour of hybridization/localization of the bulk electronic states upon both doping and depth. Core-level photoemission in the hard X-ray regime (hard X-ray photoelectron spectroscopy (HAXPES), with $h\nu > 2$ keV) supported by model calculations provides an element-specific

spectroscopic fingerprint of the electron hybridization. Distinctly different electron-screening channels exist at the surface and in the bulk, with the bulk one severely suppressed near the surface, where a stronger localization is found. Moreover, the bulk-like metallicity gradually decreases across a thickness of almost 10 unit cells (u.c.) and more than 2 u.c. (for LSMO and (Ga,Mn)As, respectively), eventually disappearing at the surface. Our findings not only deepen the comprehension of interface physics but also have a direct impact on tailor-made functionalities in new devices.

## Results

**Variable depth information via PES.** Thin films of LSMO (doping value $x = 0.33$ and $x = 0.35$, grown on LSAT(100) and SrTiO$_3$(100)) and of (Ga,Mn)As (Mn doping between 8 and 13%, grown on GaAs(100)) were measured. Figure 1 shows the Mn 2$p$ core-level spectra measured for (Ga,Mn)As (panels a and b) and LSMO (panels f and g) epitaxial films, as a function of increasing photon energy, that is, with increasing probing depth[20,21]. The lattice structures are sketched in Fig. 1 e,j, together with the depth information accessible versus photon energy. Details of growth and characterization can be found in the Methods section and Supplementary Information. Additional low binding energy (BE) features, labelled as well-screened satellites, are clearly observed for both spin–orbit partners when entering the regime of HAXPES, in agreement with previous reports[22–25]. Such features are severely reduced or absent at the lower photon energies, that is, for enhanced surface sensitivity, as highlighted in panels b and g, where the evolution of the intensity of the low-BE satellites versus photon energy is quantified by a line shape analysis (Supplementary Fig. 13). It is important to emphasize that the critical depth at which the satellite intensity appears is much larger for LSMO compared to (Ga,Mn)As, although results are obtained for the same element, Mn, and minimal differences in surface roughness are found (Supplementary Figs 10–12). We observe that the energy separation between the well-screened satellites and the main peak varies versus photon energy, as highlighted by red thick marks. We further note that in both systems the intensity increase of the low-BE satellites is gradual, with no indication of a sharp transition.

**Quantifying hybridization/localization of electronic states.** The term well-screened for the low-BE features seen in Fig. 1 is strictly linked to the screening of electrons after a core hole is created in the photoemission process. A sketch of the energy levels of ground and final states is presented in panels a and c of Fig. 2 for (Ga,Mn)As and LSMO, respectively. The BE of each possible final state depends on how effective the core hole is screened by the valence electrons, and taking into account the hybridization $V$ between valence and conduction electrons, a state can be pulled down in energy by an amount $Q$ due to the core–valence Coulomb interaction. In this condition, the core–hole potential creates an excitonic state with a hole in the 3$d$ band. In the well-screened state a valence electron, via hybridization $V$, fills this hole; in the poorly screened state, this hole stays empty.

A theoretical understanding of the satellites features in core levels of strongly correlated systems dates back several decades starting from observations in 4$d$- and 4$f$-based materials[26–28], followed by more general models for 3$d$ transition metals[29,30]. More recently, well-screened satellites have been reported as bulk-only features in core-level HAXPES spectra of 3$d$ transition metal oxides (vanadates, manganites) and magnetic diluted systems[22–25,31,32]. Following these results, improved theoretical approaches have been presented, based on multisite cluster model, Anderson impurity model and effective coupling at the Fermi

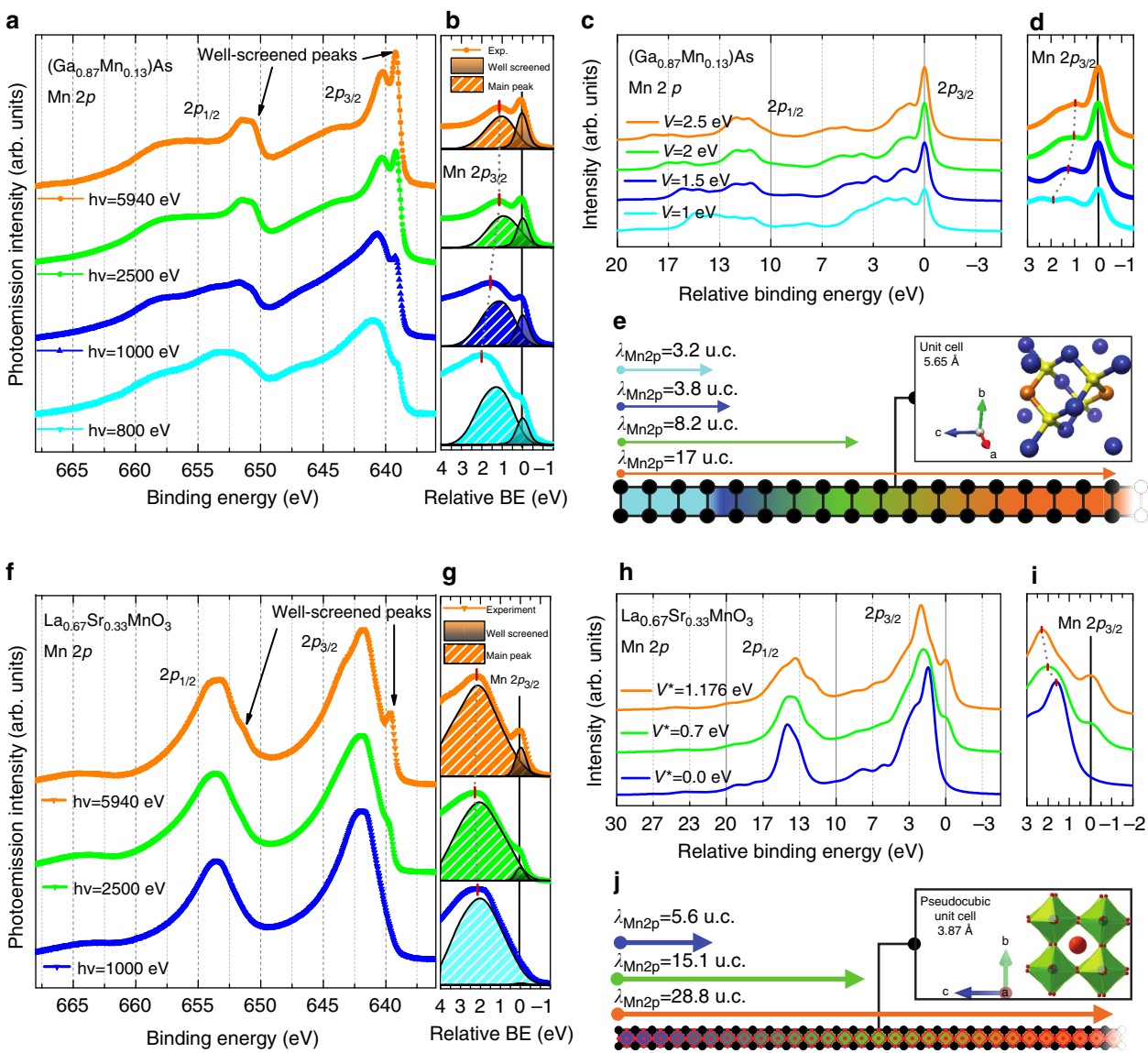

**Figure 1 | Depth-dependent PES results** (**a,b**) Photon energy-dependent Mn 2$p$ core-level spectra of (Ga,Mn)As (13% Mn-doped GaAs, measured with linear polarization at $T = 300$ K) and (**f,g**) Mn 2$p$ core-level spectra of LSMO (La$_{0.67}$Sr$_{0.33}$MnO$_3$, measured with linear polarization at $T = 210$ K). The spectra were shifted vertically for ease of comparison after integral background subtraction. The multiplet structures of the 2$p_{1/2}$ and 2$p_{3/2}$ are clearly resolved. The arrows indicate the position of the well-screened satellites for each spin–orbit partner. (**b,g**) Expanded view of the Mn 2$p_{3/2}$ peak around 640 eV BE ((**b**) (Ga,Mn)As, (**g**) LSMO). The evolution of the well-screened peak intensity is shown as a function of increasing photon energy (dotted curves are experimental spectra, filled peaks are the result of a fitting procedure described in the Supplementary Information). Note that at $hv = 1,000$ eV the well-screened intensity is clearly present in (Ga,Mn)As, while almost absent for LSMO. Calculated spectra for (Ga,Mn)As (**c**) and LSMO (**h**) from models described in the text with varying hybridization parameter $V$ and $V^*$ for (Ga,Mn)As and LSMO, respectively; the spectra are shifted vertically for clarity. (**d,i**) Close-up of the Mn 2$p_{3/2}$ peak; the thick marks indicate the different energy separations between main and satellite peak versus hybridization value $V$ and $V^*$. (**e,j**) Sketch of the depth of information and crystal structures of (Ga,Mn)As and LSMO. The length of the arrows is proportional to the inelastic mean free path $\lambda_{\text{IMFP}}$ in Mn and attenuation length, calculated as defined in refs 20,21, where the information depth corresponds to three times the inelastic mean free path $\lambda_{\text{IMFP}}$. The values of $\lambda_{\text{IMFP}}$ indicate the number of u.c. probed with the respective photon energy, identified by the colour code.

level[24,33–35]. By quantifying the variation of the hybridization parameter, the here-performed model calculations provide evidence of the link between the satellites and the metallic/insulating character of the system. Calculated spectra are shown in Fig. 1 for (Ga,Mn)As (panels c,d) and LSMO (panels h,i).

**Comparison with calculations in Mn-doped GaAs.** Photoemission spectra for the transitions $3d^n \rightarrow 2p^5 3d^n$ in (Ga,Mn)As are calculated using an Anderson impurity model,

taking into account configuration interaction in the initial and final states[33]. A scheme of the initial- and final-state configuration is presented in the insets of Fig. 2. In the calculation, $\Delta = E(d^6\underline{L}) - E(d^5)$ is the ligand-to-3$d$ charge-transfer energy, $U$ is the 3$d$–3$d$ Coulomb energy and $Q$ is the 2$p$–3$d$ Coulomb energy (we will omit $\underline{L}$ in the notation, as there presence is clear from charge neutrality). The configurations are mixed by hybridization, which is described by a parameter $V$ (see Methods). The $\Delta$, $U$ and $Q$ values give the relative energies of the average configuration as $E(3d^4) = -\Delta + U = 3$ eV,

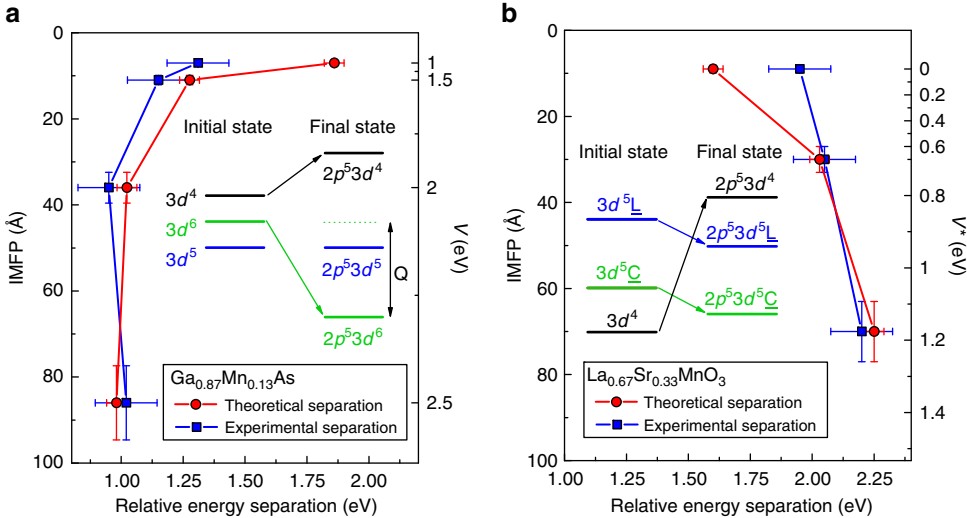

**Figure 2 | Evolution of the hybridization parameters V and V\*.** Comparison between experimental and theoretical data regarding the energy separation between well-screened and main peaks. The experimental data of (Ga,Mn)As (blue filled squares, **a**) and LSMO (blue filled squares, **b**) are plotted versus the calculated inelastic mean free path (left vertical axis) and compared with the theoretical data (red filled circles in both panels) corresponding to the value of the hybridization parameter $V$ and $V^*$ (right vertical axis in eV). Lines through the points are guides for the eye. The horizontal error is estimated as the measured experimental energy resolution, which is always larger than the positioning uncertainty of the fitting functions. The vertical error is the 10% of the IMFP value corresponding to each photon energy, according to the TPP-2M formula. The value of mean escape depth $\lambda_{IMFP}$ is calculated as the product of the inelastic mean free path and the cosine of the angle between the normal to the sample surface and the photoelectron direction ($\theta \simeq 3°$). The separation between the peaks follows the trend predicted by theory. In the insets: Schematic energy level diagram of the $3d$ configurations before hybridization in the initial and final state (in presence of a core hole), following the theoretical models described in the text. The configuration-averaged energies of the ionic configurations have been used, neglecting multiplet, crystal field, and hybridization effects in the diagram. It should be kept in mind, however, that all these effects shift and spread the levels.

$E(3d^5) = 0\,\text{eV}$ and $E(3d^6) = \Delta = 1\,\text{eV}$, and for the final state as $E(2p^53d^4) = -\Delta + U + Q = 8\,\text{eV}$, $E(2p^53d^5) = 0\,\text{eV}$ and $E(2p^53d^6) = \Delta - Q = -4\,\text{eV}$. This means that, while the ground state has primarily $d^5$ character, the lowest final state has mainly $d^6$ character. The final state $2p^53d^6$ is pulled down below $2p^53d^5$ by an energy $\sim Q$. Thus, the order of the energy levels $3d^5$ and $3d^6$ is reversed in the final state, which means that the $2p^53d^5$ is a poorly screened state and the $2p^53d^6$ a well-screened state with an extra $d$-electron. Here we adopt values $\Delta = 1\,\text{eV}$, $U = 4\,\text{eV}$, $Q = 5\,\text{eV}$ similar as for (resonant) PES[36–38] and X-ray absorption spectroscopy[39] of (Ga,Mn)As. In Fig. 1c one observes that a good agreement with the 'bulk' spectrum (orange curve in Fig. 1a) is obtained with $\Delta = 1\,\text{eV}$, $U = 4\,\text{eV}$, $Q = 5\,\text{eV}$ and $V = 2.5\,\text{eV}$, confirming the coexistence of localized and itinerant electrons in the bulk. The low-energy PES spectrum, representative of the more-localized surface region, is well reproduced with $V = 1.5$. Overall, a good agreement is found between experiment and calculation concerning the change of intensity and the relative position of satellite and main peak. The main difference with the experiment is that the calculation overestimates the intensity of the poorly screened peak. This is because the model does not include all possible additional screening channels, which can take away intensity from the screened peak, smearing out the calculated leading peak.

## Comparison with calculations in LSMO

The Mn $2p$ HAXPES spectra were calculated within the extended configuration interaction model, described in previous lines of work[24,34]. As basis states, configurations $3d^4$, $3d^5\underline{L}$ and $3d^5\underline{C}$ were used. The $3d^5\underline{C}$ represents the charge transfer (CT) between Mn $3d$ and the doping-induced coherent state at $E_F$, labelled $C$. An effective coupling parameter $V^*$ for describing the interaction strength between the Mn $3d$ and coherent state is introduced, analogous to

the Mn $3d$-O $2p$ hybridization $V$. Two parameters are varied: the CT energy between Mn $3d$ and the new $C$ states ($\Delta^*$) and the hybridization between Mn $3d$ and coherent states ($V^*$). Except for these two parameters, all other parameter values are fixed. In Fig. 1h calculated spectra for three different hybridization values $V^*$ are shown. Both the low-energy feature around 640 eV BE and a broad shoulder appearing at 644 eV with large probing depth are well reproduced by the calculations. The well-screened peak in the calculation is analysed to originate from the $2p^53d^5\underline{C}$ configuration of the final state, and increases in intensity with increasing $V^*$.

The evolution of satellite peaks in LSMO is qualitatively similar yet quantitatively different with respect to (Ga,Mn)As: a sizeable intensity of the low-BE feature in LSMO is found at much larger depth, as indicated in Fig. 1e,k. It is important to underline that the well-screened peaks correspond to the lowest-energy state. Compared to the higher-energy states, this low-energy state is less mixed with other states, which means their linewidth should be narrower. This characteristic is confirmed convincingly by the experimental results.

A further confirmation that the well-screened peak is a spectroscopic fingerprint of electron hybridization, that is, a measure of the truly bulk metallic character, is found from the comparison between the experimental data and theoretical predictions regarding the energy separation between main peak and satellite peak. In (Ga,Mn)As (Figs 1b,d and 2a), the theoretical description correctly explains the strong decrease in the energy separation for the main-to-satellite peak (red marks in Fig. 2d) going from $V = 1.0\,\text{eV}$ (surface) to $V = 2.5\,\text{eV}$ (bulk), with a larger discrepancy towards the surface limit. Although a good agreement is found between experiment and theory, theoretical data overestimate the energy separation in the range $1 < V < 2\,\text{eV}$. Calculations show that the poorly screened peak consists in fact of more than one peak. In the experimental results

we plot the energy distance to the broad main peak because the experiment cannot resolve the fine structure as theory does (see Supplementary Information). The same general agreement is also found in LSMO (Figs 1g,j and 2b) with an overall trend for increasing the energy separation between peaks when the hybridization $V^{\star}$ increases. The separation between well-screened peak ($2p^5 3d^5\underline{C}$) and main peak ($2p^5 3d^5\underline{L}$) peaks increases with $\Delta^{\star}$ and with the hybridization $V^{\star}$ between $3d^4$ and $3d^5\underline{C}$ configurations. The strength of hybridization $V^{\star}$ also influences their relative intensities. Basically, the smaller value of $\Delta^{\star}$ forms a peak at much lower BE side. Previous reports for metallic LSMO show that the separation between the main peak and the well-screened feature increases with hole-doping until $x = 0.4$, and reduces for $x = 0.55$ (ref. 22). In addition, this behaviour matches well with the metallic/magnetic character owing to the fact that, following the phase diagram, hole-doping with increasing $x$ produces a ferromagnetic phase in LSMO with larger $T_C$ and lower resistivity up to $x = 0.4$ (refs 22,40). Consistent with the decrease of the well-screened feature at low photon energies (Fig. 1f,g), a smaller peak separation suggests a smaller effective hole-doping and a lower metallicity at the surface. It is important to underline that the observed evolution of the energy separation not only confirms the validity of our approach but could also be used as a signature of the competition between localized and delocalized electronic character.

**Critical thickness of electronic hybridization**. Having ascertained that the presence and the evolution of the well-screened peaks are spectroscopic fingerprints of the electronic hybridization, we now deepen our analysis to a more quantitative aspect. Following the results of Figs 1 and 2, the attenuation of the low-BE satellites versus probing depth can be modelled by a fitting procedure described in the Supplementary Fig. 3 displays the value of the inelastic mean free path (IMFP) $\lambda_{\mathrm{IMFP}}$ at different photon energies versus the ratio between the area of the satellite peak and the total area of the Mn $2p_{3/2}$ spectrum. Here $\lambda_{\mathrm{IMFP}}$ was obtained from the TPP-2M

formula for (Ga,Mn)As and LSMO[21,41]. The curves through the points in Fig. 3 are the best fits to the data of an exponential attenuation function of the form $I(\lambda) = A\exp(-B/\lambda)$, where $A$ indicates the contribution of the satellite to the total area of the spectrum and $B$ represents the thickness of a surface layer where the screening channel associated to the extra peak is considered as totally absent, that is, a layer fully attenuating the intensity of the satellite present in the bulk. This simple model describes the attenuation of the bulk electronic hybridization when going towards the surface of the solids; hence, the thickness $B$ is a measure of the spatial extension of the modified electronic properties or, more precisely, the depth at which both hybridization and metallicity start to be modified with respect to their bulk values. A sketch of the solids with these two critical thicknesses is presented on the right (GaMnAs) and left (LSMO) sides of Fig. 3. We obtain a value of $(12 \pm 1)$ Å for (Ga,Mn)As and $(40 \pm 2)$ Å for LSMO, corresponding to more than 2 and almost 10 u.c., respectively. These values should be compared with the 10 Å-thick Mn depletion reported in (Ga,Mn)As (ref. 11), and with X-ray magnetic spectroscopy and transport results on thin LSMO films, where altered ferromagnetism and metallicity have been observed up to 30 Å of thickness from the surface[42–46].

**Strain and temperature dependence**. Strain-driven interface engineering is a relevant issue for spintronic heterostructure, and recent results have shown that the symmetry breaking induced by lattice strain influences the electron occupancy of the out-of-plane orbitals in the topmost surface layer[47]. To investigate the effect of strain on our findings, we have measured the critical thickness of bulk metallic screening using the same method of Fig. 3, on LSMO films (100 u.c. thick each) epitaxially grown on (001)-oriented $SrTiO_3$ (STO) and on $SrLaAlO_4$ (SLAO) substrates, producing, respectively, 1% tensile strain and 3% compressive strain (Supplementary Figs 1–8).

As shown in Fig. 4e, strained and unstrained films display the same main spectral features, with the exception of LSMO grown on SLAO, where a slightly broader satellite is observed. Such

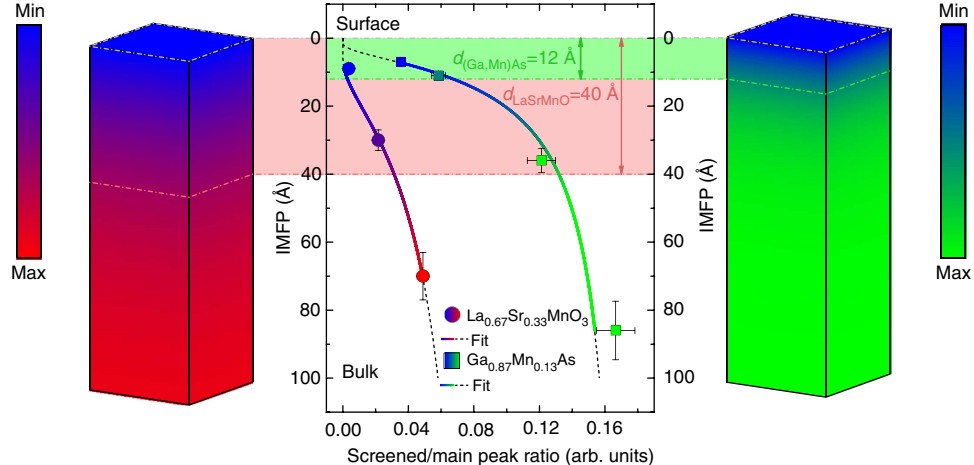

**Figure 3 | Critical thickness determination.** Evolution of the ratio between the area of the extra peak and the main peak as a function of the inelastic mean free path (filled circles for LSMO; filled squares for (Ga,Mn)As). The vertical error bar is calculated, as in Fig. 2, to be the 10% of the absolute value of the IMFP. The horizontal uncertainty is calculated by means of error propagation from the uncertainty on the parameters of the fitting functions. The solid hued lines show the curves obtained by fitting the experimental points with the function $I(\lambda) = A\exp(-B/\lambda)$. The dashed black lines show the extrapolation of the same functions outside the data range. The light green (red) shaded area shows the value corresponding to the critical thickness, that is, the value of the $B$ parameter obtained from each fit, resulting in 12 Å for (Ga,Mn)As and 40 Å for LSMO. The two blocks on left and right sides, representing respectively LSMO and (Ga,Mn)As, give a three dimensional image of the change of electron hybridization when moving from the bulk to the surface. Dark colours correspond to a zone where hybridization is reduced with respect to its bulk value. The filling is hued from blue to green (red) in the vertical direction according to the fitted function for (Ga,Mn)As (LSMO), with blue corresponding to zero and green (red) corresponding to the maximum value assumed by the function in the range 0–100 Å.

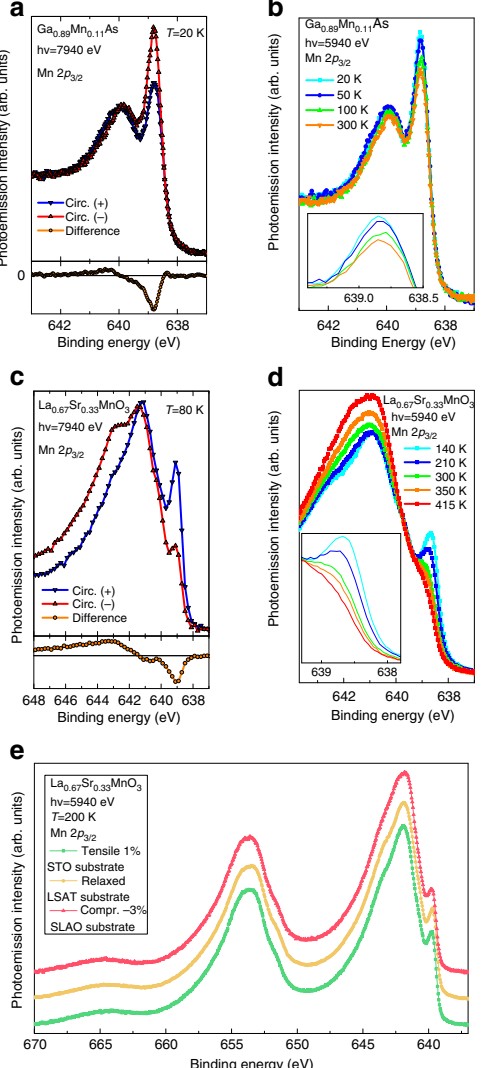

**Figure 4 | Magnetic dichroism and Mn 2p versus temperature and strain.**
(**a**,**c**) Mn 2p core-level spectra measured with left- and right-circularly
polarized X-rays in LSMO (**c**) and (Ga,Mn)As (**a**). Differences (open
circles) are shown at the bottom of the panels. Note that the largest
magnetic signal is located at the energy position of the well-screened
satellites. (**b**,**d**) Temperature dependence of the Mn $2p_{3/2}$ core-level
spectra. In LSMO (**d**), a shoulder is still visible above $T_C$ (345 K), while the
intensity of the well-screened satellites, responsible of the large dichroism
observed, is totally suppressed. In (Ga,Mn)As, (**b**), the intensity variation of
the well-screened satellite is highlighted in the inset. The maximum
increase is observed when crossing $T_C = 70$ K. (**e**) Mn 2p core level of
LSMO thin films (100 u.c.) grown on three different substrates: $SrTiO_3$
(STO), $(LaAlO_3)_{0.3}(Sr_2TaAlO_6)_{0.7}$ (LSAT), and $SrLaAlO_4$ (SLAO), measured
at $hv = 5940$ eV and T = 200 K. Due to the lattice mismatch, STO induces a
1% tensile strain and SLAO a 3% compressive one. Films grown on LSAT are
considered as relaxed. No significant difference is observed in their line
shape, except a small decrease in intensity in the case of the compressive
strain (red spectrum). Calculated critical thicknesses, using the same
method of Fig. 3, give $41 \pm 7$ Å in LSMO/SLAO, while no difference within
the error bar is found in the case of LSMO/STO.

difference is attributed to the lower Curie temperature of strained
films, as explained in the next section. The critical screening
thickness obtained from the application of the fitting procedure is
$41 \pm 7$ Å in LSMO/SLAO, while no difference within error bars is
found between the tensile-strained LSMO/STO films and the

unstrained ones grown on LSAT. These results show that,
although sensitive to the change of electronic hybridization across
the entire thickness of the film, core-level photoemission does not
provide a direct probe of orbital occupancy and/or symmetry
breaking at the surface as the X-ray absorption technique used in
ref. 47, suggesting that the evolution of the bulk metallic
screening across the thickness of the sample could be attributed
to dimensionality effects.

We now turn to the influence of temperature on the magnetic
properties. Chemical sensitive magnetic information can be
obtained by detecting the Mn 2p core-level spectra using left-
and-right circularly polarized X-rays (magnetic circular dichro-
ism in HAXPES), as shown in panel a Fig. 4a for (Ga,Mn)As and
panel c for LSMO, respectively. A large difference, the magnetic
circular dichroism, is observed with the well-known down-up
($2p_{3/2}$), up-down ($2p_{1/2}$) line shape[32,48,49]. It is important to
emphasize that, in both systems, the largest magnetic signal is
observed at the energy position of the well-screened satellites,
confirming the major role of delocalized electrons in establishing
the mechanism of ferromagnetism[12,23,24,32]. Furthermore,
Fig. 4b,d shows that the temperature also influences the relative
intensities of the well-screened and poorly screened peaks. In
(Ga,Mn)As, a clear change in the relative intensity of the main
and satellite peaks is visible when crossing $T_C = 70$ K, as
previously reported[23]. As for LSMO, the inset of panel d reveals
a fine structure of the well-screened satellites, where: (i) a weak
shoulder persists above $T_C$ (345 K), corresponding to a multiplet
structure not directly linked to ferromagnetic properties and (ii) a
large decrease of spectral weight is observed, connected to the
change of the magnetic order parameter, confirming previous
results on $La_{1-x}Ba_xMnO_3$ (ref. 50). In this regime of doping, the
bilayered manganites have a concomitant metal–insulator and
ferromagnetic–paramagnetic transition as a function of
temperature and the well-screened satellite disappears above
$T_C$ (ref. 25). It should be noted that the decrease of the well-
screened satellite intensity follows a bulk-like behaviour and does
not display the well-known rapid decrease usually observed in
surface-sensitive results[51]. Although the largest dichroism,
corresponding to the presence of a long-range magnetization, is
observed where delocalized states are, a contribution to
magnetism is also arising from the rest of the spectrum, that is,
localized or poorly screened part. This further corroborates the
fact that the magnetism is reduced in absence of the extra-peak
screening channel, although not suppressed.

## Discussion

We provide clear evidence from the study of two paradigmatic
spintronics materials that the electronic hybridization strength
varies significantly from its bulk value when approaching the
surface layers. The dimensionally dependent role of (de)localized
electrons is revealed, affecting both the ferromagnetic and
metallic behaviour. Concerning the spatial extension of the
altered electronic properties, the interplay between carrier
localization and metallicity in (Ga,Mn)As is restricted to the
near-surface region ($\sim 10$ Å) in agreement with the observed loss
of long-range magnetization and Mn depletion leading to a
superparamagnetic-like response[11]. As for LSMO, previous
reports suggested the existence of decoupled magnetic and
metallic critical thicknesses due to localization effects in thin
films, with a larger extension (up to 30 Å) of the metallic
thickness[42,44,52–54]. Our results corroborate this hypothesis,
giving a quantitative estimate of the crossover thickness range
for a film to fully develop bulk-like hybridization values. It is
important to emphasize that the spectroscopic fingerprints we
have used and modelled are representative of a bulk-only

character: the strong attenuation of the well-screened feature measured close to the surface is directly connected to an altered metallicity, yet not suggesting that this metallicity is completely suppressed. Moreover, we have shown that both localized and delocalized electrons contribute to both electronic and magnetic character, opening the perspective of a direct comparison between the critical thickness for the electronic structure with the critical thickness of the magnetic properties. Finally, our data provide evidence that a difference in the overall film properties will be always present due to the reduced dimensionality, hence due to the modified hybridization, especially when the film is not thick enough and/or at interfaces. The depth extension of the surface/interface electronic charge distribution, and its control via atomic-precision growth techniques, will conceivably open alternative routes for engineering complex heterostructures.

## Methods

**Growth.** $La_{0.65}Sr_{0.35}MnO_3$ (LSMO) thin films were grown by molecular beam epitaxy in a dedicated chamber located at the APE beamline (NFFA facility, Trieste, Italy). LSMO films of different doping and thickness have been grown. Samples used in the present report are grown on $La_{0.18}Sr_{0.82}Al_{0.59}Ta_{0.41}O_3$ (LSAT), that is, strain relaxed, and have a thickness of 100 u.c. ($\approx 400$ Å), with the $c$ axis oriented perpendicularly to the surface. Growth and characterization of strained films are described in Supplementary Material. The ferromagnetic (Ga,Mn)As films (Mn doping level between 6 and 13%) were grown by molecular beam epitaxy on GaAs substrates using a modified Veeco Gen II system at the University of Regensburg, Germany, with a Mn doping range between 6 and 13%. Samples measured in present report have 200 Å of thickness and 12.5% of doping, unless reported otherwise. (Ga,Mn)As films were measured as-grown, in order to avoid segregation of Mn towards the surface after post-annealing treatment. Details of structural and magnetic characterization can be found in the Supplementary Material.

**HAXPES.** Photon energy- and temperature-dependent HAXPES experiments were performed at the I09 beamline at Diamond Light Source (Didcot, UK). The beamline's end station is equipped with a SCIENTA EW-4000 electron energy analyser, mounted with the lens axis perpendicular to the X-ray beam. A grazing incidence geometry (that is, normal photoelectron emission) has been used, that is, 3° angle between the X-ray beam and the surface plane, with an angular and energy resolution better than 0.2° and 20 meV, respectively, resulting in a $30 \times 250\,\mu m^2$ beam footprint on the sample. The position of the Fermi level $E_F$ and the overall energy resolution have been estimated by measuring the Fermi edge of a poly-crystalline Au foil in thermal and electric contacts with the samples. The overall energy resolution (analyser + beamline) was kept below 250 meV over the entire photon energy range. Magnetic dichroism in HAXPES spectra was acquired at BL19LXU at Spring-8, Japan, equipped with a SCIENTA R4000-10 kV electron energy analyser at grazing incidence geometry ($>4°$ angle between the X-ray beam and the surface plane) and a spotsize of $40 \times 500\,\mu m^2$ on the sample. The overall energy resolution (analyser + beamline) was kept below 400 meV at the selected photon energy.

**Calculations.** *(GaMn)As.* Photoemission spectra for the transitions $3d^n \rightarrow 2p^5 3d^n$ in (Ga,Mn)As are calculated using an Anderson impurity model, taking into account configuration interaction in the initial and final states[26]. The wave functions are a coherent sum over $d^n$, $d^{n+1}\underline{L}$, $d^{n+2}\underline{L}^2,...$ configurations, where $\underline{L}$ denotes a hole in the ligand orbitals. The average energies of the initial and final state configurations are taken as points on a parabola, $E(d^n) = n(\Delta - 5U) + \frac{1}{2}n(n-1)U$ and $E(2p^5 d^n) = E(2p^5) + E(d^n) - nQ$, respectively, where $\Delta = E(d^6\underline{L}) - E(d^5)$ is the ligand-to-3d charge-transfer energy, $U$ is the $3d$–$3d$ Coulomb energy and $Q$ is the $2p$–$3d$ Coulomb energy. The configurations are mixed by hybridization with a parameter $V = \langle d|H|\underline{L}\rangle$.

The Hamiltonians, $H$, for the initial and final states of the Mn atoms are calculated using Cowan's code[55], including tetrahedral crystal-field symmetry ($10Dq = -0.5$ eV), spin–orbit and multiplet structure but neglecting band structure dispersion. Wave functions are calculated in intermediate coupling using the atomic Hartree–Fock approximation with relativistic corrections and by reducing the Slater parameters to 80% to account for intra-atomic correlation effects. No coherent state near the Fermi level has been included. When $V$ is larger, there will be more weight of the $3d^6$ configuration in the initial state, resulting in a more intense 'satellite' peak at ~4 eV lower BE than the main peak, giving a $d$ count of 5.2 and ground state of 0.3% $d^3$, 10.0% $d^4$, 60.8% $d^5$, 26.5% $d^6$ and 2.8% $d^7$.

*LSMO.* A modified Anderson impurity model was used with six configurations as basis states: $3d^4$, $3d^5\underline{L}$ and $3d^5\underline{C}$. The $3d^5\underline{C}$ represents the CT between Mn $3d$ and doping-induced coherent state at $E_F$, labelled $C$. The $V(\Gamma)$, $Udd$ and $-Udc$ are the hybridization between the TM $3d$ and the ligand states, the on-site repulsive Coulomb interaction between TM $3d$ states and the attractive core–hole potential,

respectively. The standard Hamiltonian $H$mult describes the intra-atomic multiplet coupling between TM $3d$ states and that between TM $3d$ and TM $2p$ states. The spin–orbit interactions for TM $2p$ and $3d$ states are also included. A further term $V^*$, is introduced to account for the 'coherent' screening channel at the Fermi level[22,40], with an effective coupling parameter accounting for interaction strength with $3d$ states. The CT energy for such additional state is $\Delta^*$. We used a value of $U = 5.1$ Vh for the on-site Coulomb repulsion, $Q = 5.4$ eV for the attractive core–hole potential, $\Delta^* = 4.5$ eV for the CT energy, $10Dq = 1.5$ eV for the crystal field and $V(e_g) = 2.94$ eV. The reduction factors of $V$ are $R_c = 0.8$, $R_v = 0.9$.

**Data availability.** The data that support the findings of this study are available from the corresponding author on request.

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

## Acknowledgements

This work has been partly performed in the framework of the nanoscience foundry and fine analysis (NFFA-MIUR Italy) facility.

## Author contributions

G.P., T.P. and F.B. conceived the experiment and wrote the paper with contributions from G.v.d.L., M.T., C.H.B. and G. R. All authors discussed the results, commented manuscript and prepared written contributions. F.M.G., P.G. and A.Y.P. grew the films, and R.C. and M.C. characterized samples. G.v.d.L. and M.T. performed the calculations. F.B., V.L., B.G. T.-L.L., C.S., M.O., H.D., A.R. and D.J.P. performed synchrotron radiation experiments and analysed the data.

## Additional information

**Competing interests:** The authors declare no competing financial interests.

