## [Peer Review File · Nature Communications]

Reviewers' Comments:

Reviewer #1 (Remarks to the Author):

Understanding the crossover regime between bulk and surface properties is important for the interpretation and prediction of novel electronic and magnetic phenomena in new material phases and in low dimensional structures. This is particularly relevant in the field of interface-driven spintronics, where a few tuning parameters (doping, material thickness, etc.) can be used to influence macroscopic properties, including charge and spin transport. Developing a quantitative model of how translational symmetry breaking at a crystal surface affects the electronic coupling on and between atomic sites relative to the bulk structure is crucial for engineering new devices for charge and spin manipulation.

The paper of Pincelli et al. addresses this problem using a very focussed and elegant approach. By combining X-ray photoemission spectroscopy and Anderson impurity model calculations, the authors show how the degree of orbital hybridisation at different atomic sites evolves as a crystalline surface is approached in two typical spintronics systems, LSMO and (Ga, Mn)As. They identify specific depth-dependent features in the photoemission spectra, the so-called well screened satellite peaks, and monitor their relative emission intensities as a function of photon energy, i.e. of X-ray penetration in the sample. Well screened satellite peaks are bulk-only features and they mark the onset of strong orbital hybridisation and truly bulk metallic character. Calculations are then used to model the observed spectrum, by considering, in the Anderson impurity model approximation, interaction between a selected number of orbital configurations, corresponding to potential hybridisation channels. A hybridisation parameter is obtained from the calculations, which quantifies the orbital mixing. Effectively, this approach allows one to monitor, potentially quantitatively, how orbital hybridisation changes moving from the surface, through the surface layer, to reach the bulk, and how ferromagnetism and metallic behaviour are affected by the change in hybridisation. The results for the two systems studied indicate that deviations from bulk electronic properties may still be detectable deep in the sample (up to 30 angstrom from the surface in the case of LSMO) and, furthermore, that the transition from surface to bulk properties occurs gradually, as indicated by the smooth evolution of the satellite peak intensities with penetration length.

This is novel, sound and far reaching work of potential major relevance in spintronics research. The main conclusions are well justified and clear, and they are consistent with previous work on the materials examined. It is intriguing to speculate whether this novel approach might be extended to wider classes of systems, and whether it may be complemented with both experimental (e.g. electron microscopy) and computational (ab initio and post-DFT) techniques to achieve a more complete understanding of the interplay between bulk crystal structure, surface reconstruction and electronic/magnetic properties of spintronics materials. The level of detail in the paper is appropriate and the information provided in the Methods section should be sufficient to make the work reproducible. The quality of the presentation and clarity are superb. I believe the novelty, importance and potential impact of the work make the paper suitable for publication in Nature Communications.

Reviewer #2 (Remarks to the Author):

The manuscript “Quantifying the critical thickness...” by Pincelli et al reports investigation of local electronic structure of Mn atoms in two paradigmatic spintronics materials LSMO and Ga(Mn)As using hard-X-ray photoelectron spectroscopy (HAXPES). Variation of photon energy and thus probing depth of this experiment has allowed identification of the critical thickness where crossover between the surface and bulk electron hybridization and correlation takes place. Analysis of the experimental data is supported by calculations based on the Anderson impurity model (AIM). The manuscript is clear and well written.

My concern is however the novelty of the presented material. Extensive HAXPES measurements on LSMO, interpreting the well-screened peak on the low-BE side Mn 2p core level as signature of the bulk electronic structure and supported by cluster calculations, have already been reported in a few works of the Spring-8 group (the author’s Refs. 20, Horiba et al., JES 144–147 (2004) 557). The only their difference to the present manuscript is the use of angle rather than photon energy dependence to evaluate the thickness dependence. Temperature dependence of the Mn 2p signal has also been reported in the previous works. Regarding (Ga,Mn)As, HAXPES experiments on this material scrutinizing the well-screened Mn 2p peak and supported by AIM calculations have also been previously reported (Ref. 21 and Fujii et al, PRL 111 (2013) 097201), with the only difference being that the present manuscript adds two other photon energies. HAXPES magnetic circular dichroism and temperature dependence, although only for two temperatures, were also reported in the two above papers as well as in the review in Ref. 30. Novelities of the present manuscript are however (1) quantitative evaluation of the critical thickness, although essentially from the same HAXPES dataset, and, to the best of my knowledge, (2) HAXPES magnetic circular dichroism for LSMO.

Regarding more specific points of the manuscript, the authors should estimate what was the X-ray penetration depth for the 3-deg incidence angle and photon energies used to acquire the HAXPES data. Is it indeed much smaller than the photoelectron escape depth and therefore does not affect the critical thickness determination? Furthermore, the authors should definitely mention the previous work of Pescuera et al. (Nature Comm. 3 (2012) 1189) that discusses thickness dependent orbital filling in LSMO investigated by linear dichroism of XAS. This work suggests strain dependence of electronic structure in thin films of LSMO which can possibly be avoided using LSMO monocrystals (Lev et al. PRL 114 (2015) 237601). The authors may comment whether the strain effects were at play for their LSMO and (Ga,Mn)As samples and whether they might affect the determined critical thickness.

According to the editorial policy of Nature Communications, the referees only access the scientific novelty and quality of the manuscripts. Having outlined these aspects of the present manuscript, I am leaving the publication recommendation to the full editor's discretion.

Reviewer #3 (Remarks to the Author):

The manuscript of Pincelli et al. presents core-level X-ray photoemission spectroscopy (PES) for $\text{La}_{1-x}\text{Sr}_x\text{MnO}_3$ and $\text{Ga}_{1-x}\text{Mn}_x\text{As}$ for several X-ray photon energies $\hbar\nu$ and temperatures. The spectra are analyzed theoretically within a cluster Anderson impurity model. It is concluded that the magnitude of both the parameter V^* in $\text{La}_{1-x}\text{Sr}_x\text{MnO}_3$ (describing hybridization of d-states with a coherent state at the Fermi energy in Mott-Hubbard insulators) and the parameter V in $\text{Ga}_{1-x}\text{Mn}_x\text{As}$ (describing hybridization of Mn d-states and p-states of anion ligands in charge transfer insulators) get smaller when $\hbar\nu$ and, thus, the probing length by PES, decreases.

While the manuscript after revision could be published in another journal, I do not regard it as suitable for publication in Nature Communications. The arguments are as follows.

1. As far as I can see, in the case $\text{La}_{1-x}\text{Sr}_x\text{MnO}_3$, both the data and the theoretical model employed are similar (if not identical) to those in ref. 20, so that the novelty can be questioned. In particular, both in ref. 20 and in the present manuscript a systematic reduction of the shoulder structure at low $\hbar\nu$ is taken as an evidence that this feature has a surface electronic structure component.
2. In the case of $\text{Ga}_{1-x}\text{Mn}_x\text{As}$, the data cover a more extensive $\hbar\nu$ range but again both the data and the theoretical model are similar to those in ref. 45, so that once more the novelty can be questioned. Furthermore, in this case, in contrast to $\text{La}_{1-x}\text{Sr}_x\text{MnO}_3$, the explanation of the data in terms of reduced V for lower $\hbar\nu$ (cf. figs. 1b and 1d) does not look convincing.

3. Despite that the manuscript is co-authored by experience authors, a number of sentences and references appear pretentious and/or unfitting. Some examples:

(i) The first two sentences and the associated refs. 1 - 4.

(ii) Ref. 7 was invalidated by later papers of the same group and by experimental results in Ref. 36.

(iii) Ref. 8 does not contain information quoted in the manuscript. Moreover it was invalidated by Wang et al., Phys. Rev. B 87, 121301(R) (2013).

Reply to Reviewers

'Quantifying the critical thickness of electron hybridization in spintronics materials', by T. Pincelli et al.

Nat. Commun. manuscript number NCOMMS-16-27276A

Reviewer #1

We thank the reviewer for her/his positive comments and for the appreciation of our work.

Reviewer #2

We thank the reviewer for her/his comments and criticisms.

Reviewer. *My concern is however the novelty of the presented material. Extensive HAXPES measurements on LSMO, interpreting the well-screened peak on the low-BE side Mn 2p core level as signature of the bulk electronic structure and supported by cluster calculations, have already been reported in a few works of the SPring-8 group (the author's Refs. 20, Horiba et al., JES 144–147 (2004) 557). The only their difference to the present manuscript is the use of angle rather than photon energy dependence to evaluate the thickness dependence. Temperature dependence of the Mn 2p signal has also been reported in the previous works. Regarding (Ga,Mn)As, HAXPES experiments on this material scrutinizing the well-screened Mn 2p peak and supported by AIM calculations have also been previously reported (Ref. 21 and Fujii et al, PRL 111 (2013) 097201), with the only difference being that the present manuscript adds two other photon energies. HAXPES magnetic circular dichroism and temperature dependence, although only for two temperatures, were also reported in the two above papers as well as in the review in Ref. 30.*

We agree with the reviewer that the presence of bulk-only satellites in HAXPES core levels has already (and extensively) been reported by many groups, including the co-authors of the present paper. However, we do think that the novelty of our paper builds on the fact that we **have quantified for the first time** the change in electron hybridization when going from the bulk to the surface (as also the reviewer points out in her/his comments). Our present observations and analysis encompass and combine previous results and allow displaying the full picture. We have provided a method **to determine a critical thickness, potentially useful for each material**, below which the localization becomes dominant and above which the bulk band structure is recovered. This has large implications for interface physics and spintronics in general. This aspect has been clearly underlined by Reviewer #1.

Moreover, our results go well beyond just measuring the critical thicknesses, namely, (a) it requires a number of unit cells for the bulk properties to fully develop, which is mostly a dimensionality issue and not the result of reconstruction or non-stoichiometry of a surface layer, (b) it may be a general property of spintronic materials, and (c) the combination of magnetically-sensitive and electronic-sensitive results including the temperature dependence allows us to directly compare the critical thickness for the electronic structure with the critical thickness of the magnetic properties. The electron transport in manganites is coupled to the ferromagnetism: our results suggest that these two thicknesses are related, if not similar in length. We have modified accordingly our discussion in the main paper, and we thank the reviewer for having noticed this important point.

Moreover, we present for the first time accurate measurements showing that the relative positions of the satellite and main peak vary (see fig. 1b and 1d for (Ga,Mn)As, fig. 1g and 1j for LSMO, and fig. 2) when passing from localized to delocalized character, in full agreement with Anderson impurity model calculations where the hybridization parameter is varied.

Reviewer *Novelties of the present manuscript are however (1) quantitative evaluation of the critical thickness, although essentially from the same HAXPES dataset, and, to the best of my knowledge, (2) HAXPES magnetic circular dichroism for LSMO.*

We also agree with the reviewer that magnetic dichroism curves of (Ga,Mn)As have already been reported in previous papers. Our idea was to compare results with LSMO, and clarify the relationship between satellite peaks and magnetism, thus corroborating our picture where localized and delocalized electrons at the same time contribute to both electronic and magnetic properties in spintronics materials. We would like to underline that the temperature dependence of satellite peak in LSMO, measured in great detail (i.e., revealing that it is composed by one peak following the temperature dependence and a shoulder not depending on temperature), is original and new, to the best of our knowledge.

Reviewer. *Regarding more specific points of the manuscript, the authors should estimate what was the X-ray penetration depth for the 3-deg incidence angle and photon energies used to acquire the HAXPES data. Is it indeed much smaller than the photoelectron escape depth and therefore does not affect the critical thickness determination?*

We thank the reviewer for her/his comment. We agree that the grazing incidence geometry might be a relevant parameter affecting our measurements, because when the measurements are carried out at very low incidence angles the x-ray penetration depth might decrease considerably to values of the same order as the photoelectron escape depth [C.S. Fadley, *Progress in Surface Science*, **16**, 275 (1984)]. For fixed grazing incidence angle, this effect is more relevant for soft rather than hard x-ray photon energies. We have added a section with text, figures and calculations where we show that in our experiment the penetration depth of the incident x-rays is always larger than the Mn 2p photoelectron escape depth, by about one order of magnitude. This result emerges through comparing the attenuation length of the incident x-rays and the Mn 2p photoelectrons for each photon energy used in our experiment.

Reviewer. *Furthermore, the authors should definitely mention the previous work of Pescuera et al. (Nature Comm. 3 (2012) 1189) that discusses thickness dependent orbital filling in LSMO investigated by linear dichroism of XAS. This work suggests strain dependence of electronic structure in thin films of LSMO which can possibly be avoided using LSMO monocrystals (Lev et al. PRL 114 (2015) 237601). The authors may comment whether the strain effects were at play for their LSMO and (Ga,Mn)As samples and whether they might affect the determined critical thickness.*

We thank the reviewer for this comment, as it has helped us to improve our paper.

As for (Ga,Mn)As, we did not explore strain-related issues: for Mn diluted in the GaAs lattice, the generation of significant strain implies a level of control in the growth procedure that we are not able to reach.

We have measured the HAXPES Mn 2p core level of LSMO thin films grown on different substrates, inducing both tensile and compressive strain. We have added text and figures in the supplementary materials

regarding the growth and characterization, and added one panel and a section in the main text describing our findings.

We have measured LSMO films 100 u.c. thick, namely on SrTiO₃ (STO) (1% tensile strain), (LaAlO₃)_{0.3}(Sr₂TaAlO₆)_{0.7} (LSAT) (almost relaxed), and SrLaAlO₄ (SLAO) (-3% compressive strain). We observe slight variations of the lineshape in LSMO/SLAO, and almost no difference for LSMO/STO. In LSMO/SLAO, applying the fitting routine for the estimation of the critical bulk metallic screening we obtain 41 ± 7 Å, i.e., a minimal difference with respect to the relaxed films.

The paper mentioned by the reviewer provides information on the orbital occupancy showing that:

- The topmost layer of the film has a natural unbalance in the occupation of the $3z^2-r^2$ and x^2-y^2 due to the lack of apical oxygen and consequently a reduction of the electrostatic repulsion along the z direction.
- The natural unbalance can be compensated, and even reversed, by the application of strain, which lifts the degeneracy between the $3z^2-r^2$ and x^2-y^2 orbitals.

Moreover, measurements have been performed well above the Curie temperature, in order to suppress the magnetic effects in the XLD signal.

In our study, the HAXPES technique does not provide enough sensitivity to probe the surface layer, nor core level spectra measurement hold direct information on orbital occupancy, as could be the case measuring valence band photoemission. The absence of changes in bulk metallic screening in the presence of strain-induced rearrangement of the orbital occupation suggests that the value of the critical thickness is related mainly to dimensionality effects, and is less influenced by electronic/orbital reconstructions or stoichiometry. Concerning the work by Lev et al., the photon energy used in their valence band experiments (and consequently the kinetic energy that defines the bulk sensitivity) is <1000 eV, and we are not able to compare our data with the ones of Lev et al. Also in this case, valence band HAXPES should be performed to evaluate the possible influence of strain. We thank the referee for pointing out what we now believe it is a very relevant point in our discussion, allowing us to further increase the novelty of our results.

Reviewer #3

We thank the reviewer for the useful comments and criticisms.

Reviewer 1. *As far as I can see, in the case $La_{1-x}Sr_xMnO_3$, both the data and the theoretical model employed are similar (if not identical) to those in ref. 20, so that the novelty can be questioned. In particular, both in ref. 20 and in the present manuscript a systematic reduction of the shoulder structure at low is taken as an evidence that this feature has a surface electronic structure component.*

The Anderson impurity model (AIM) is a well-established model providing a reliable description for the localized vs delocalized behaviour. It is often used for charge-transfer compounds, including insulators and semiconductors. It would break down for fully metallic (i.e., itinerant) systems. We agree with the reviewer that in ref. 20 (now ref. [22]), as well as in many other published papers since then, the analysis of the satellite features in core level photoemission has led to the conclusion that the surface electronic properties are different compared to the bulk properties. Although the electron hopping process is known to be reduced in

the vicinity of the surface (which is often recognized as an important effect), in the present work we are able to quantify this effect, showing that it is larger than what intuitively might have been expected, with clear implications for interface-based effects often found in spintronics applications.

Moreover, for the first time, accurate measurements with good energy resolution made it here possible to highlight that the relative positions of the satellite and main peak vary (see fig. 1b and 1d for (Ga,Mn)As, fig. 1g and 1j for LSMO, and fig. 2) when passing from localized to delocalized character, in full agreement with AIM calculations where the hybridization parameter is varied.

In Ref. 20, and previously published works, no attempt has been made to quantify the localized vs delocalized behaviour, nor to study the fine details of both theory and experiment with respect to the hybridization parameter.

Both these issues are new, and represent for the first time a solid base to build a quantitative analysis of HAXPES spectra as well as a precise estimate of the critical electronic (and possibly magnetic) thickness.

Reviewer 2. *"In the case of $Ga_{1-x}Mn_xAs$, the data cover a more extensive $h\nu$ range but again both the data and the theoretical model are similar to those in ref. 45"*

Although the theoretical model is similar (see our reply above to comment 1), ref. 45 (now ref. [49]) does not present the photon energy dependence of HAXPES spectra (only doping dependent spectra at fixed photon energy) nor temperature dependence of the dichroic signal (another difference in ref. 45 is that linear dichroism is presented, not circular). The authors did not address the issue of the thickness dependence or the critical thickness at which the electronic structure is modified.

Reviewer 2. *Furthermore, in this case, in contrast to $La_{1-x}Sr_xMnO_3$, the explanation of the data in terms of reduced V for lower $h\nu$ (cf. figs. 1b and 1d) does not look convincing.*

Despite its limitations, in many cases reported in the literature, the AIM describes the localized vs delocalized behaviour rather well. In fig. 1c and 1d of our paper it can be seen that for increasing hybridization V , the intensity of the well-screened peak increases. The same effect is seen in fig. 1a for the experimental data with increasing photon energy. We agree with the reviewer that, while the trend is the same, in the calculation the intensity of the well-screened peak is more pronounced than in the experiment. The main reason for this is that the model does not include all possible additional screening channels, which can take away additional intensity from the screened peak. Including other channels will smear out the leading peak in the calculation, giving thus the impression of a better agreement, yet masking the main results of our analysis: i) The intensity of the satellite peak changes when passing from surface to bulk environment (i.e., having less and more hybridization, respectively) and (ii) the relative position of the satellite and main peak varies (results in fig. 2) in both calculation and experiment: this is a signature, not previously reported, of the validity of our model.

Reviewer 3. *Despite that the manuscript is co-authored by experience authors, a number of sentences and references appear pretentious and/or unfitting. Some examples:*

Consiglio Nazionale delle Ricerche

Istituto Officina dei Materiali
Area Science Park - Basovizza, Ed. MM
Strada Statale 14 km 163,5 - 34149 Trieste
<http://www.iom.cnr.it> info@iom.cnr.it

Trieste, Cagliari, Grenoble, Perugia

(i) *The first two sentences and the associated refs. 1 - 4.*

(ii) *Ref. 7 was invalidated by later papers of the same group and by experimental results in Ref. 36.*

(iii) *Ref. 8 does not contain information quoted in the manuscript. Moreover it was invalidated by Wang et al., Phys. Rev. B 87, 121301(R) (2013).*

(i) We have added a reference (Jungwirth T. et al., Rev. Mod. Phys. **86**, 855 (2014), now ref. [5]), as a more specific one for (Ga,Mn)As. We have modified the first sentences; we did not have the intention to be pretentious. In the first two phrases of the paper we tried to introduce the general argument of spintronics and interfaces, but we are open to specific suggestions.

(ii) Ref. [7] has been updated as suggested by the reviewer.

(iii) We have corrected a mistake in citing ref. [8]. We thank the reviewer for noticing our mistake. We added a further reference as suggested by the reviewer: Wang M. et al., Phys. Rev. B **87**, 121301(R) (2013), now ref. [10].

Sincerely,

Giancarlo Panaccione

Consiglio Nazionale delle Ricerche - Istituto Officina dei Materiali

Sede di Istituto Trieste: Area Science Park–Basovizza, Ed. MM Strada Statale 14 Km 163.5 – 34149 Trieste, Italy, ☎ (+39)040 3756411, fax (+39) 040 226767

UOS Trieste: Via Bonomea 265, 34136 Trieste, Italy, ☎ (+39)0403787443, fax (+39)0403787528

UOS Cagliari: Dipartimento di Fisica, Cittadella Universitaria – 09042 Monserrato, Cagliari, Italy ☎ +39-0706754893, fax (+39)0706754892

Sede di lavoro OGG Grenoble: c/o ESRF, 6 rue J. Horowitz, BP220 F-38043 Grenoble Cedex 9 ☎ +33 (0)476 882857, fax +33 (0)476 882855

Sede di lavoro Perugia: Dipartimento di Fisica – Università di Perugia, Via A. Pascoli, 06123 Perugia, Italy ☎ (+39)0755853060, fax (+39)0755852737

Unità trasversale di supporto: Corso Perrone 24, 16152 Genova, ☎ (+39)0106598750, fax –(+39)0106506302

Partita IVA IT 02118311006 – C.F. 80054330586

Reviewers' Comments:

Reviewer #2:

Remarks to the Author:

The revised version of the manuscript is a significant improvement. The authors responded to my concerns regarding the X-ray penetration depth with a careful analysis in the Supplementary, and to those regarding the strain effects with additional measurements on LSMO grown on different substrates providing both tensile and compressive strain.

Still there are nevertheless my concerns regarding novelty of the most of the presented experimental data (apart from the less significant part on temperature dependence of the satellite in LSMO). However, the strength of the manuscript is rather in systematization of the known data. I am leaving to the editors the final decision whether this kind of research is suitable for Nature Communications. However, I am personally leaning to recommend the publication because the problem of surface vs bulk effects in electronic structure of two prototype spintronics materials is certainly interesting and motivating for a wide scientific community.